## [Transparent Peer Review file · Nature Communications]

Transcriptional competence defines the heterochromatin nucleating potential of isolated MSR units

Corresponding Author: Dr Thomas Jenuwein

Version 0:

Reviewer comments:

Reviewer #1

(Remarks to the Author)

Review of "Isolated DNA repeat units nucleate heterochromatin through Integrator-coupled transcriptional attenuation" by Lo et al.

Heterochromatin domains are specialized regions of the genome marked by methylation of histone 3 lysine 9 (H3K9me3). These domains play crucial roles in maintaining genome stability by repressing transcription at specific locations and silencing repetitive elements. Paradoxically, the establishment of heterochromatin requires transcriptional activity, whereby the transcripts are targeted by RNA processing machinery leading to recruitment of Suv39h histone methylation enzyme activity to form a heterochromatin domain. Exactly how cells distinguish transcripts that should trigger heterochromatin formation from those that are actively transcribed remains an important area of investigation. Here the authors use Major Satellite Repeat (MSR) transcripts in mouse cells as a tool for discovering the required features, as these repeats are dynamically regulated during early development and found aberrantly expressed in some cancers, suggesting they play key roles in establishing silenced heterochromatin domains. By inserting isolated MSR repeat units into an inert intergenic chromosome region, they find that multiple copies (3 or 9) are sufficient to produce transcripts and H3K9me3 enrichment, while other repeat elements like LINE sequences are not. Moreover, they report that the full length MSR consensus sequence, when inserted at 9 copies, is sufficient to establish an extended domain of H3K9me3 enrichment. They find that MSR transcripts lack typical mRNA features (5' cap and polyA tail), and that they are processed by the Integrator complex associated with RNA Polymerase II. The authors conclude that approximately 15% of mouse MSR repeats maintain transcriptional competence and that their regulation by the Integrator complex is a key mechanism for distinguishing transcripts that should trigger heterochromatin formation.

Critique: The results demonstrating that the insertion of three or more transcriptionally competent copies of major satellite repeats, which produce non-mRNA transcripts, can induce heterochromatin formation at otherwise inert genomic regions are highly compelling. Additionally, the authors show that MSR-derived transcripts are attenuated by the Integrator complex. These findings support a transcription-coupled model for heterochromatin assembly at MSR units in pericentric regions, which will be of significant interest to the field. I recommend the publication of this exciting study in Nature Communications. The following minor suggestions can be addressed through textual revisions.

1. Previous studies have demonstrated that HP1 can bind to transcripts and promote heterochromatin assembly through its interaction with histone methyltransferase enzymes. The authors should consider discussing the possibility that HP1 may mediate MSR-driven heterochromatin assembly at the ectopic site.
2. Although the correlation between the presence of MSR transcripts and heterochromatin formation is compelling, it remains possible that transcription factors binding directly to MSR units could recruit heterochromatin factors independently of transcript production. This possibility can be acknowledged in the discussion.
3. RNA processing and transcription termination factors, such as CPSF, have been shown to cooperate in promoting heterochromatin assembly in *S. pombe* (see PMID: 22144463, 24210919, 31269446). These papers that first identified the involvement of RNA processing factors in heterochromatin assembly should be cited, along with more recent work by Khanduja and colleagues (ref 23). Additionally, it is noteworthy that Integrator components such as INTS9 and INTS11 are homologous to CPSF subunits

(Remarks on code availability)

Reviewer #2

(Remarks to the Author)

Lo et. al, provide mechanistic insight into how specific major satellite repeat (MSR) elements can initiate heterochromatin formation in the mouse genome. Using a reductionist approach, the authors show that only a subset of transcriptionally competent MSR units — particularly when inserted as tandem repeats — can establish de novo heterochromatin at an otherwise inert, repeat- and gene-free genomic locus. These "heterochromatin nucleating" MSRs promote a compact chromatin structure involving HP1a recruitment and histone H1 incorporation, independent of changes in H3K9me3 levels. The authors also show that these MSRs are regulated by the Integrator complex, particularly that of its transcription. Integrator depletion leads to increased transcription without altering the repressive H3K9me3 mark but reduces HP1a binding, suggesting a transcription-coupled mechanism of chromatin regulation. The authors further show that the non-coding MSR transcripts are stabilized by R-loops and non-B form DNA topology, and act through Integrator and Suv39h to reinforce heterochromatin architecture.

Altogether, the study links repeat DNA sequence, RNA processing, and chromatin remodeling in a unified framework and highlights a novel RNA quality control role for the Integrator complex in shaping heterochromatin in repeat-rich mammalian genomes. We found this manuscript presents an exciting conceptual advance and is worthy of publication- congratulations to the authors on a nice piece of thought provoking work!

We recommend the following clarifications or added experimental support to solidify the conclusions. We encourage the authors to consider addressing before publishing the work.

1) The authors discuss the influence of repeat copy numbers on heterochromatinization. Extended Data Fig. 2a presents endogenous MSR loci with ~1.2–1.5 copies, whereas synthetic insertions in Fig. 2c suggest that ≥ 3 tandem MSR copies are required for robust heterochromatin nucleation. It remains unclear whether copy number alone is sufficient or if specific sequence or structural features of the repeats also contribute. The authors should clarify whether both length and sequence fidelity are essential parameters, and if there is a defined threshold for chromatin remodeling. They can validate this by inserting the exact copy number as in the endogenous MSR loci and then compare that with ≥ 3 tandem MSR copies (Figure 2C).

2) In figure 2f the authors show higher RNA Pol II enrichment at MSR3 than MSR9, but it is unclear whether this correlates with heterochromatin protein recruitment at MSR9. The authors do not show HP1 α or histone H1 enrichment at MSR9. Inclusion of these data would help evaluate whether RNAPII levels inversely correlate with chromatin compaction and provide mechanistic insight into differential repeat behavior.

3) Despite comparable H3K9me3 levels at both MSR3 and MSR9 (Fig. 2c), RNA Pol II occupancy differs, raising questions about whether H3K9me3 correlates with transcriptional repression in these loci. The authors should discuss whether this points to a decoupling of H3K9me3 from RNAPII exclusion, or whether there are additional factors that influence transcriptional activity.

4) In figure 3C the authors quantify the amount of T7 RNAP at MSR3 and MSR9, however, it is not understood why the two graphs are separated with different sets of clones (i.e. WT, HB, L5'UTR3, CMV3, MSR3 and WT, HB MSR1, MSR3, MSR9). Can they be joined in one graph?

5) In figure 4B the authors suggest that the transcription from the MSR is bidirectional, but why there is no elongating polymerase at the MSR3 insert either at 5' end or at 3' end.

6) The authors attribute MSR transcription to Pol II and its regulation by the Integrator complex. However, repetitive elements can also be transcribed by Pol III. Given the repetitive DNA context and non-coding nature of MSRs, have the authors ruled out a contribution from Pol III or other RNA polymerases? Including or discussing this possibility would strengthen the mechanistic claims.

7) In figure 5B the authors show that in absence of integrator, the H3K9me3 is still present but the HP1 recruitment decreases. It would be interesting to assess what happens to other heterochromatin marks like H3K27me3 (for which the last author is world famous).

8) The authors show different clones of the MSR copies in the figures, but this is not mentioned in the figure legends or text. The authors should specify in the figure legend that the denotation is specifying a particular clone.

9) The integrin and R-DNA loop formation is slightly disjointed in the mechanistic angle, and in the current state slightly detracts from a very nice story. We suggest either removing it, or buttressing it with additional data and controls. For instance, did the authors also check what happens if R-DNA loops are inhibited by RNase H treatment?

10) Early in the manuscript, transcription factor (TF) binding to MSRs is mentioned, yet it is not revisited or integrated into the final model. Do any specific TFs modulate MSR activity or contribute to heterochromatin formation? If such factors were identified in the screen or bioinformatics analysis, their relevance could be updated- this angle is obviously fascinating and under-explored in the field. However, it might be better suited to the Discussion/Future plans section.

(Remarks on code availability)

Reviewer #3

(Remarks to the Author)

(Remarks on code availability)

Reviewer #4

(Remarks to the Author)

(Remarks on code availability)

Reviewer #5

(Remarks to the Author)

This manuscript by Lo and colleagues investigates heterochromatin formation and silencing of major satellite repeats (MSR). Utilising a reductionist approach by inserting individual MSR copies either singly or in 2, 3 or 9 copies at an inert, ectopic locus in mouse ES cells, they find that only transcriptionally competent repeats when inserted in at least 3 tandem copies lead to heterochromatin establishment. This is also associated with the low-level expression of small, bi-directional and chromatin-associated RNAs lacking a cap, as well as formation of RNA:DNA hybrids. Finally, they find that the Integrator complex is required for repression of transcription from both ectopic and endogenous major satellite repeats.

This is a very well-designed and well-executed study. The data is of high quality with all the appropriate controls and replicates demonstrating robustness of the results. The Figures are extremely well presented. Overall, the conclusions are convincingly demonstrated, and I am happy to support publication of this study in Nature Communications. The findings are also of high interest to the study of heterochromatin formation and transcriptional silencing and the use of this reductionist model is likely to provide important additional advances in the future. I just have a few minor points detailed below:

1. Supplementary Figure 1 should be moved to the Extended Data.

2. Extended Data Fig. 7: the top panel doesn't have a panel letter.
3. Please show evidence for depletion of INTS11 protein with the degron system.
4. Figure 5c and Extended Data Fig.8. It is unclear whether the heatmaps are ordered according to sequence identity or signal enrichment.
5. Figure 6e is referred to as 6d right panel and Figure 6d as left panel.
6. Figure 7 what are the red and blue loops?

(Remarks on code availability)

Version 1:

Reviewer comments:

Reviewer #1

(Remarks to the Author)

The authors have addressed all my concerns. I have no further comments

(Remarks on code availability)

Reviewer #3

(Remarks to the Author)

(Remarks on code availability)

Reviewer #4

(Remarks to the Author)

(Remarks on code availability)

Reviewer #5

(Remarks to the Author)

I had only very minor comments in my original review, which the authors have now addressed in full.

(Remarks on code availability)

REVIEWER COMMENTS

Reviewer #1 (Remarks to the Author):

Review of “Isolated DNA repeat units nucleate heterochromatin through Integrator-coupled transcriptional attenuation” by Lo et al.

Heterochromatin domains are specialized regions of the genome marked by methylation of histone 3 lysine 9 (H3K9me3). These domains play crucial roles in maintaining genome stability by repressing transcription at specific locations and silencing repetitive elements. Paradoxically, the establishment of heterochromatin requires transcriptional activity, whereby the transcripts are targeted by RNA processing machinery leading to recruitment of Suv39h histone methylation enzyme activity to form a heterochromatin domain. Exactly how cells distinguish transcripts that should trigger heterochromatin formation from those that are actively transcribed remains an important area of investigation. Here the authors use Major Satellite Repeat (MSR) transcripts in mouse cells as a tool for discovering the required features, as these repeats are dynamically regulated during early development and found aberrantly expressed in some cancers, suggesting they play key roles in establishing silenced heterochromatin domains. By inserting isolated MSR repeat units into an inert intergenic chromosome region, they find that multiple copies (3 or 9) are sufficient to produce transcripts and H3K9me3 enrichment, while other repeat elements like LINE sequences are not. Moreover, they report that the full length MSR consensus sequence, when inserted at 9 copies, is sufficient to establish an extended domain of H3K9me3 enrichment. They find that MSR transcripts lack typical mRNA features (5' cap and polyA tail), and that they are processed by the Integrator complex associated with RNA Polymerase II. The authors conclude that approximately 15% of mouse MSR repeats maintain transcriptional competence and that their regulation by the Integrator complex is a key mechanism for distinguishing transcripts that should trigger heterochromatin formation.

Critique: The results demonstrating that the insertion of three or more transcriptionally competent copies of major satellite repeats, which produce non-mRNA transcripts, can induce heterochromatin formation at otherwise inert genomic regions are highly compelling. Additionally, the authors show that MSR-derived transcripts are attenuated by the Integrator complex. These findings support a transcription-coupled model for heterochromatin assembly at MSR units in pericentric regions, which will be of significant interest to the field. I recommend the publication of this exciting study in Nature Communications. The following minor suggestions can be addressed through textual revisions.

1. Previous studies have demonstrated that HP1 can bind to transcripts and promote heterochromatin assembly through its interaction with histone methyltransferase enzymes. The authors should consider discussing the possibility that HP1 may mediate MSR-driven heterochromatin assembly at the ectopic site.

The function of HP1 and the Suv39h-HP1 pathway is described in the introduction. To what extent HP1 association contributes to MSR-instructed heterochromatin formation at the integration site is currently not resolved (lines 418-420).

2. Although the correlation between the presence of MSR transcripts and heterochromatin formation is compelling, it remains possible that transcription factors binding directly to MSR

units could recruit heterochromatin factors independently of transcript production. This possibility can be acknowledged in the discussion.

Whether transcription factor binding could recruit heterochromatin components independently of MSR transcript production is currently not resolved and this is acknowledged in the discussion (lines 418-420).

3. RNA processing and transcription termination factors, such as CPSF, have been shown to cooperate in promoting heterochromatin assembly in *S. pombe* (see PMID: 22144463, 24210919, 31269446). These papers that first identified the involvement of RNA processing factors in heterochromatin assembly should be cited, along with more recent work by Khanduja and colleagues (ref 23). Additionally, it is noteworthy that Integrator components such as INTS9 and INTS11 are homologous to CPSF subunits

*The papers on RNA quality control factors (Lee et al., 2013 and Vo et al., 2019) have now been cited (line 69). It is also discussed that the cleavage and polyadenylation specificity factors (CPSF) in *S.pombe* are orthologous to the INTS9/INTS11 subunits of Integrator (line 438-441).*

Reviewer #2 (Remarks to the Author):

Lo et. al, provide mechanistic insight into how specific major satellite repeat (MSR) elements can initiate heterochromatin formation in the mouse genome. Using a reductionist approach, the authors show that only a subset of transcriptionally competent MSR units — particularly when inserted as tandem repeats — can establish de novo heterochromatin at an otherwise inert, repeat- and gene-free genomic locus. These "heterochromatin nucleating" MSRs promote a compact chromatin structure involving HP1a recruitment and histone H1 incorporation, independent of changes in H3K9me3 levels. The authors also show that these MSRs are regulated by the Integrator complex, particularly that of its transcription. Integrator depletion leads to increased transcription without altering the repressive H3K9me3 mark but reduces HP1a binding, suggesting a transcription-coupled mechanism of chromatin regulation. The authors further show that the non-coding MSR transcripts are stabilized by R-loops and non-B form DNA topology, and act through Integrator and Suv39h to reinforce heterochromatin architecture.

Altogether, the study links repeat DNA sequence, RNA processing, and chromatin remodeling in a unified framework and highlights a novel RNA quality control role for the Integrator complex in shaping heterochromatin in repeat-rich mammalian genomes. We found this manuscript presents an exciting conceptual advance and is worthy of publication-congratulations to the authors on a nice piece of thought provoking work!

We recommend the following clarifications or added experimental support to solidify the conclusions. We encourage the authors to consider addressing before publishing the work.

1) The authors discuss the influence of repeat copy numbers on heterochromatinization. Extended Data Fig. 2a presents endogenous MSR loci with ~1.2–1.5 copies, whereas synthetic insertions in Fig. 2c suggest that ≥ 3 tandem MSR copies are required for robust heterochromatin nucleation. It remains unclear whether copy number alone is sufficient or if specific sequence or structural features of the repeats also contribute. The authors should clarify whether both length and sequence fidelity are essential parameters, and if there is a defined threshold for chromatin remodeling. They can validate this by inserting the exact copy number as in the endogenous MSR loci and then compare that with ≥ 3 tandem MSR copies (Figure 2C).

The analysis of intergenic MSR variants has been the starting point for this study. As described in the text and shown in supplementary Figures 1 and 2, intergenic MSR variants are not isolated but are found in the immediate vicinity of other repeat elements and it was therefore unclear whether they have an intrinsic potential to induce H3K9me3. To solve this question, we developed the reductionist approach by inserting isolated MSR copies into the inert Chr2/116 region. We compared single and three-copy insertions of an intact (act-9/35) and of a permuted (per-3/99) MSR variant and also generated MSR1, MSR2, MSR3 and MSR9 insertions of the MSR consensus sequence. We feel that the comparative data for MSR1, MSR2, MSR3, MSR9 and between act-9/35 and per-3/99 MSR variant insertions are comprehensive and provide compelling evidence to support a model in which topological alterations require a minimum sequence of three copy insertions (Figures 1c, 2c, 6d, 6e, 6f). In addition, induction of H3K9me3 is also dependent on three, but not one or two copy MSR insertions (Figures 1c, 2c and supplementary Figure 13) and on the presence of non-mutated transcription factor binding sites (Figure 1c, supplementary Figure 2b and supplementary Figure 4). Thus, both sequence length (i.e. minimum of three copy insertions) and sequence identity (presence of non-mutated transcription factor binding sites) are essential to instruct MSR-based heterochromatin formation. We have clearly detailed this in the discussion (lines 406-429) and in an extended description of the model Figure 7.

2) In figure 2f the authors show higher RNA Pol II enrichment at MSR3 than MSR9, but it is unclear whether this correlates with heterochromatin protein recruitment at MSR9. The authors do not show HP1 α or histone H1 enrichment at MSR9. Inclusion of these data would help evaluate whether RNAPII levels inversely correlate with chromatin compaction and provide mechanistic insight into differential repeat behavior.

We have added new data on histone H1 enrichment for the MSR9 insertion (new Figure 3d, right panel). The MSR9 insertion has increased signals for histone H1 incorporation as compared to the MSR3 insertion and can help to explain that chromatin compaction reduces RNAPII recruitment (lines 235-240).

3) Despite comparable H3K9me3 levels at both MSR3 and MSR9 (Fig. 2c), RNA Pol II occupancy differs, raising questions about whether H3K9me3 correlates with transcriptional repression in these loci. The authors should discuss whether this points to a decoupling of H3K9me3 from RNAPII exclusion, or whether there are additional factors that influence transcriptional activity.

The ChIP-qPCR only detects signals that are internal (i.e. three copies) to the construct-specific T7 and T3 primers. We cannot analyze ChIP-qPCR across the nine copies (around

2 kb of chromatin fragments) of the MSR9 insertions. As described above, MSR9 insertions have higher histone H1 incorporation and reduced RNAPII recruitment. Decreased levels of MSR9-derived transcripts are therefore likely caused by reduced RNAPII recruitment (lines 235-240).

4) In figure 3C the authors quantify the amount of T7 RNAP at MSR3 and MSR9, however, it is not understood why the two graphs are separated with different sets of clones (i.e. WT, HB, L5'UTR3, CMV3, MSR3 and WT, HB MSR1, MSR3, MSR9). Can they be joined in one graph?

These analyses compare two different kinds of insertions. The first comparison is between three copy insertions of L5'UTR3, CMV3 and MSR3 (Figure 3c) and the second comparison is between MSR1, MSR3 and MSR9 (Figure 3d, left panel). We prefer to maintain these separate graphs, also because we now added the new data on histone H1 enrichment for the MSR1, MSR3 and MSR9 insertions (new Figure 3d, right panel).

5) In figure 4B the authors suggest that the transcription from the MSR is bidirectional, but why there is no elongating polymerase at the MSR3 insert either at 5'end or at 3' end.

MSR-derived transcripts have a short length that can only extend into the 5'P1 and 3'P1 positions (Figure 4c). Since the Integrator is known to regulate RNAPII pausing and promoter-proximal termination of RNAPII-derived transcripts (line 289-291), the absence of signals for elongating RNAPII at the 5'P0 and 3'P0 positions would be consistent with Integrator-mediated attenuation of RNAPII function at the MSR3 insertions.

6) The authors attribute MSR transcription to Pol II and its regulation by the Integrator complex. However, repetitive elements can also be transcribed by Pol III. Given the repetitive DNA context and non-coding nature of MSRs, have the authors ruled out a contribution from Pol III or other RNA polymerases? Including or discussing this possibility would strengthen the mechanistic claims.

We have added new data on RNAPI, RNAPII and RNAPIII enrichment across MSR1, MSR3 and MSR9 insertions (new supplementary Figure 6). MSR instructed transcription is specifically regulated by RNAPII, as no signals for RNAPI or RNAPIII could be detected across MSR insertions (lines 193-195).

7) In figure 5B the authors show that in absence of integrator, the H3K9me3 is still present but the HP1 recruitment decreases. It would be interesting to assess what happens to other heterochromatin marks like H3K27me3 (for which the last author is world famous).

Polycomb (i.e. H3K27me3) repressive chromatin can partially compensate H3K9me3 in constitutive heterochromatin, if the Suv39h enzymes are depleted. Since our analysis is focused on MSR-instructed heterochromatin in wild-type and not in Suv39h double-null mouse ES cells, we have not performed ChIP-qPCR for H3K27me3.

8) The authors show different clones of the MSR copies in the figures, but this is not mentioned in the figure legends or text. The authors should specify in the figure legend that the denotation is specifying a particular clone.

We have amended the Figure legends to indicate that ES cell clone identities are specified by numbers.

9) The integrin and R-DNA loop formation is slightly disjointed in the mechanistic angle, and in the current state slightly detracts from a very nice story. We suggest either removing it, or buttressing it with additional data and controls. For instance, did the authors also check what happens if R-DNA loops are inhibited by RNase H treatment?

The topological alterations of multi-copy MSR insertions are important parameters to facilitate RNAPII engagement and allow bi-directional transcription, which is then attenuated by Integrator. We have strengthened the data on topological alterations of multi-copy MSR insertions by including topoisomerase 2 (Top2) inhibition (new Figure 6f). Only for MSR3 and MSR9 insertions, but not for MSR1 and MSR2 insertions did we detect significantly elevated 5' and 3' Chr2/116 transcripts following exposure to etoposide, a known Top2 poison. These data validate a topological response of isolated MSR repeat units and are consistent with a recent publication showing that endogenous MSR transcripts become elevated after Top2 inhibition (Fuhrmann et al., 2025).

10) Early in the manuscript, transcription factor (TF) binding to MSRs is mentioned, yet it is not revisited or integrated into the final model. Do any specific TFs modulate MSR activity or contribute to heterochromatin formation? If such factors were identified in the screen or bioinformatics analysis, their relevance could be updated- this angle is obviously fascinating and under-explored in the field. However, it might be better suited to the Discussion/Future plans section.

We have referred to transcription factor binding at MSR sequences and transcription factor-based models for heterochromatin formation in the introduction. Since the MSR DNA consensus sequence is full of embedded transcription factor binding sites and there are > 700 Zn-finger and homeo domain TF in the mouse proteome, many of those could bind to MSR sequences not only in mouse ES cells but also in other cell types. The importance of transcription factor binding in further promoting RNAPII engagement and in instructing MSR heterochromatin is clearly described in the text (lines 149-151 and 424-427) and explained in the extended Figure legend of model Figure 7.

Reviewer #3 (Remarks to the Author):

Reviewer #4 (Remarks to the Author):

Reviewer #5 (Remarks to the Author):

This manuscript by Lo and colleagues investigates heterochromatin formation and silencing of major satellite repeats (MSR). Utilising a reductionist approach by inserting individual MSR copies either singly or in 2, 3 or 9 copies at an inert, ectopic locus in mouse ES cells, they find that only transcriptionally competent repeats when inserted in at least 3 tandem copies lead to heterochromatin establishment. This is also associated with the low-level expression of small, bi-directional and chromatin-associated RNAs lacking a cap, as well as formation of RNA:DNA hybrids. Finally, they find that the Integrator complex is required for repression of transcription from both ectopic and endogenous major satellite repeats.

This is a very well-designed and well-executed study. The data is of high quality with all the appropriate controls and replicates demonstrating robustness of the results. The Figures are extremely well presented. Overall, the conclusions are convincingly demonstrated, and I am happy to support publication of this study in Nature Communications. The findings are also of high interest to the study of heterochromatin formation and transcriptional silencing and the use of this reductionist model is likely to provide important additional advances in the future. I just have a few minor points detailed below:

1. Supplementary Figure 1 should be moved to the Extended Data.

All supplementary Figures are now listed as supplementary Figures 1-13.

2. Extended Data Fig. 7: the top panel doesn't have a panel letter.

The Western blot for INTS11 siRNA depletion in supplementary Figure 9 is indicated as being shown on top.

3. Please show evidence for depletion of INTS11 protein with the degron system.

We have added the Western blot to confirm depletion of INTS11-degron in Figure 5c.

4. Figure 5c and Extended Data Fig.8. It is unclear whether the heatmaps are ordered according to sequence identity or signal enrichment.

We have specified this in the Figure legends (Figure 5c and supplementary Figure 11): MSR sequences were classified into two groups based on sequence identity (95-100% conservation to the MSR DNA consensus sequence and below 95% conservation to the MSR DNA consensus sequence). Within each group, the signals were ordered according to their level of enrichment.

5. Figure 6e is referred to as 6d right panel and Figure 6d as left panel.

This has been rectified.

6. Figure 7 what are the red and blue loops?

We have extended the Figure legend of model Figure 7 and detail all indicated structures, symbols and abbreviations.